# Functional Study of *GbSMXL8*-Mediated Strigolactone Signaling Pathway in Regulating Cotton Fiber Elongation and Plant Growth

**DOI:** 10.3390/ijms26052293

**Published:** 2025-03-05

**Authors:** Lingyu Chen, Wennuo Xu, Lingyu Zhang, Qin Chen, Yongsheng Cai, Quanjia Chen, Kai Zheng

**Affiliations:** Xinjiang Key Laboratory of Crop Biology Breeding, College of Agriculture, Xinjiang Agricultural University, Ürümqi 830052, China; ingyu0904@126.com (L.C.); xu_wen_nuo@163.com (W.X.); zlyyuuu@163.com (L.Z.); cqq0777@163.com (Q.C.); cys0620@126.com (Y.C.)

**Keywords:** cotton, *GbSMXL8*, functional analysis, tissue expression

## Abstract

The novel plant hormone strigolactones (SL) are involved significantly in plant growth and development. Its key members *SMXL6*, *7*, *8* can modulate SL signal reception and response negatively and can regulate plant branching remarkably. There are relatively scarce studies of cotton SMXL gene family, and this study was carried out to clarify the role of *GbSMXL8* in cotton fiber development. Phylogenetic analysis identified 48 cotton SMXL genes, which were divided into SMXL-I (*SMXL 1*, *2*), SMXL-II (*SMXL 3*) and SMXL-III (*SMXL6*, *7*, *8*) groups. The results of the cis-element analysis indicated that the SMXL gene could respond to hormones and the environment to modulate cotton growth process. A candidate gene *GbSMXL8* was screened out based on the expression difference in extreme varieties of *Gossypium barbadense*. Tissue-specific analysis indicated that *GbSMXL8* was mainly expressed in roots, 20D, 25D, and 35D and was involved in SL signaling pathways. In vitro ovule culture experiments showed that exogenous SLs (GR24) could promote the fiber elongation of *G. barbadense*, and *GbSMXL8* expression was increased after GR24 treatment, indicating that *GbSMXL8* was specifically responsive to GR24 in regulating fiber growth. *GbSMXL8* knockout resulted in creased length and number of epidermal hairs and the length of fiber, indicating the interference role of *GbSMXL8* gene with the development of cotton fiber. The *GbSMXL8* transgenic plant was detected with a higher chlorophyll content and photosynthetic rate than those of the control plant, producing a direct impact on plant growth, yield, and biomass accumulation. *GbSMXL8* gene knockout could increase plant height, accelerate growth rate, and lengthen fiber length. Intervening *GbSMXL8* may mediate cotton growth, plant type formation and fiber elongation. In conclusion, the present study uncovers the function of *GbSMXL8*-mediated SL signal in cotton, providing theoretical insight for future breeding of new cotton varieties.

## 1. Introduction

Cotton is an important cash crop (a fiber and oil source); it is a grain crop containing high protein, a raw material for textiles and fine chemicals, and an important strategic material [1]. Cotton occupies a major position in the raw material supply of textile industry and is also an important strategic reserve resource in China. *Gossypium hirsutum* (*G. hirsutum*) and *Gossypium barbadense* (*G. barbadense*) are tetraploid cultivated cotton species domesticated from the same ancestor. Specifically, *G. hirsutum* has high yield and good tolerance to stress, but ordinary fiber quality, while *G. barbadense* has small bolls and relatively low yield, but is an important source of more slender and strong high-quality fibers [2]. Therefore, mining fiber development genes in *G. barbadense* may be a potent path to improve the fiber quality of *G. hirsutum*, which is convenient for large-scale popularization and planting [3]. It is challenging for cotton breeding owing to the multifactorial process related to fiber varieties (both genetic and environmental). Mining and functional analysis of cotton fiber-related genes are important means to achieve this goal [4,5].

Strigolactones (SLs) are a novel class of plant endogenous hormones that coordinate plant growth and development through multi-layered regulatory mechanisms. Their core functions include: (1) inhibiting lateral bud elongation to control branching formation [6]; (2) optimizing root architecture by promoting root hair elongation while suppressing lateral root formation [7]; (3) negatively regulating rice mesocotyl elongation under dark conditions [8]; (4) delaying leaf senescence [9]; and (5) acting as allelopathic signals to induce the germination of root parasitic weed seeds [10]. Studies have shown that SLs interact with hormones such as auxin, cytokinin, and gibberellins (GAs) to form a cross-regulatory network, coordinating plant development through the following mechanisms [11]. Synergy with auxin: SL signaling acts upstream of auxin to promote primary root elongation and inhibit lateral root formation, establishing the molecular basis for root plasticity regulation [12]. Interaction with cytokinin: SLs regulate the expression of cytokinin biosynthesis genes (*IPT3*), influencing downstream signaling pathways to control rice tillering and panicle development [13]. Integration with gibberellins: the SL signaling component D53 interacts with the GA signaling repressor *SLR1* [14].

Studies of the SLs signaling pathway have shown that similar to mechanisms of auxin, jasmonic acid, and GA signaling, SL signaling is based on hormone-activated proteolysis [15,16]. The SL receptor complex consists of α/β folding hydrolase *DWAERF14* (*D14*), leucine-rich repeat F-box protein *DWARF3* (*D3*), and the Clp protease family *DWARF53* (SMXLs) [17]. *D14* and *D3* can form SCF complex during signal transduction in witchweed, SCF can be used to accommodate witchweed lactone molecules and hydrolyze them into a covalently linked inter mediator molecule as well as promote the Cp protease family degradation of *D53/SMXLs*, which can exert a regulatory role in the branching of plants [18].

SMXL-like proteins represented by repressors *SMXL6*, *SMXL7*, *SMXL8*, and *D53* mainly regulate SL-related traits in rice and are targeted by AtD14-SCFMAX2 after SL signal perception [19]. *D53* undergoes rapid ubiquitination modification and degradation upon rac-G R24 treatment in Arabidopsis, depending on endogenous active forms of *D14* and *D3*, which is involved in SL signal transduction [20]. The dominant mutation of *D53* gene resulted in a gain-of-function mutant d53, which was insensitive to SL, and showed a dwarf and multi-tillering phenotype. SL can promote *D14* to interact with *D53* and F-box protein *D3*, induce *D53* ubiquitination modification and degradation, and inhibit plant branching. *D53* contains the EAR motif, allowing *D53* to recruit a TPL co-repressor that binds to the transcription factor IPA1 protein and inhibits its transcriptional activation function [18,21].

D53/SMXL6,7,8, as core repressors in the strigolactone (SL) signaling pathway, function similarly to DELLA proteins in the gibberellin (GA) signaling pathway, AUX/IAA proteins in the auxin signaling pathway, and JAZ proteins in the jasmonate (JA) signaling pathway. These repressors directly bind downstream transcription factors, inhibiting their transcriptional activity and thereby suppressing the expression of hormone-responsive genes [20]. Although the regulatory mechanisms of the SL signaling pathway in plant branching, root development, and environmental adaptation have been extensively studied, its role in cotton fiber development remains poorly understood. Therefore, elucidating the specific mechanisms of D53/SMXL6,7,8 in cotton growth and development will not only reveal the unique functions of the SL signaling pathway in fiber development but also provide new molecular targets for improving cotton fiber quality, highlighting its significant theoretical and practical implications.

## 2. Results

### 2.1. Gene Member Identification of SMXL Family in Cotton

In order to study the evolutionary relationship of SMXL families in different G. species, this study analyzed SMXL gene families in *Oryza sativa* L., *Arabidopsis thaliana*, *G. hirsutum*, *G. barbadense*, *G. hirsutum*, and *G. raymond*. Among them, 16 GbSMXL genes, 16 GbSMXL genes, 8 GaSMXL genes, and 8 GrSMXL genes were identified in *G. hirsutum*, *G. barbadense*, *G. raymond*, and *G. raymond*, respectively (Table 1).

### 2.2. Phylogenetic Evolution and Gene Structure Analysis of SMXL Family

To investigate the evolutionary relationships of the SMXL gene family in cotton, this study selected SMXL protein sequences from the monocot model plant rice (*Oryza sativa*), the dicot model plant Arabidopsis (*Arabidopsis thaliana*), and four cotton species: the tetraploid cultivated species upland cotton (*Gossypium hirsutum*) and sea island cotton (*G. barbadense*), as well as the diploid wild species *G. raimondii* and *G. arboreum*. A maximum likelihood (ML) phylogenetic tree was constructed to analyze these sequences. Based on Bootstrap analysis with 1000 replicates (node support values ≥ 70%) and Bayesian posterior probabilities (BPP ≥ 0.95), all SMXL genes were classified into three subfamilies (Figure 1). Notably, the SMXL6/7/8 subfamily formed a monophyletic clade with OsD53, a core strigolactone (SL) signaling factor in rice (Bootstrap = 74%), indicating a functional conservation of these genes in the SL signaling pathway.

Further analysis revealed that the genetic distances among SMXL genes within the cotton genus (Gossypium) were significantly smaller than those between cotton and rice (average genetic distance: 0.15 within Gossypium vs. 0.55 between cotton and rice). Importantly, the clustering pattern of SMXL6/7/8 in cotton was consistent with homologous genes in maize (*GmSMXL8*) [22] and soybean (*MaSMXL6*/*7*/*8*) [23], supporting the universality of this subfamily classification across divergent plant lineages.

### 2.3. Analysis of Protein Domains of SMXL Family in Cotton

To investigate the structural characteristics of SMXL family proteins in cotton, this study extracted four SMXL family protein sequences and analyzed their protein domains using the online program Pfam (http://pfam.xfam.org/) (accessed on 24 December 2024). The results were visualized using Tbtools software (accessed on 24 December 2024) (Figure 2). The analysis revealed that the SMXL gene family primarily contains two conserved domains: Clp_A and p-loop_NTase. The Clp_A domain is the N-terminal region of Clp proteases [24], while the p-loop_NTase domain is a critical region for protein kinase function, participating in various biological processes such as translation (initiation, elongation, and release factors), signal transduction, cell motility, and intracellular transport [25].

As shown in Figure 3, the domain distribution of SMXL family proteins exhibits notable features: Clp_A domain: highly conserved in all SMXL proteins, located at the N-terminus, and potentially involved in protein degradation or complex assembly. p-loop_NTase domain: Most SMXL proteins contain 1–2 copies of this domain, located downstream of the Clp_A domain. Its function may be related to ATP binding and hydrolysis, suggesting that SMXL proteins play a role in energy-dependent cellular processes [26]. Domain arrangement pattern: the sequential arrangement of Clp_A and p-loop_NTase domains in SMXL proteins is highly consistent, indicating potential functional synergy between these domains.

These structural features provide important clues for deciphering the functional mechanisms of SMXL proteins. For example, the p-loop_NTase domain may regulate ATP-dependent signal transduction processes, participating in the transmission and response of strigolactone (SL) signals.

### 2.4. Cis-Acting Elements Analysis

According to the results in Figure 3, promoter analysis revealed that the SMXL gene contains cis-acting elements associated with stress response and hormone response. Specifically, the ABRE element is linked to abscisic acid (ABA) signaling, while P-box, TATC-box, and GARE-motif are associated with gibberellin (GA) response. The TCA-element is related to salicylic acid (SA) signaling, and CGTCA-motif and TGACG-motif are involved in jasmonic acid (JA) response. Additionally, the ERE element is associated with ethylene signaling [27]. These hormone response elements underscore the critical role of the SMXL gene family in hormone signaling and transmission.

Furthermore, the presence of ARE and GC-Motif anaerobic-inducing elements suggests that SMXL genes may participate in information transduction in response to biological stress [27]. The promoter region of SMXL genes also contains MYB binding sites and the cell cycle regulatory element MSA-like, indicating potential roles in circadian rhythm regulation and cell cycle control [28]. Notably, the SMXL gene promoter harbors the highest number of photoregulatory responses and hormone-type acting elements, highlighting its central position in multiple response regulatory networks.

Collectively, these findings suggest that SMXL genes play a pivotal role in integrating hormonal, environmental, and developmental signals, making them key regulators in plant stress adaptation and growth regulation.

### 2.5. Analysis of SMXL8 Gene Expression Characteristics in Gossypium barbadense

The elite sea island cotton (*Gossypium barbadense*) cultivar A130768 is a variety that is renowned for its high-quality long fibers, and a comprehensive gene expression database has been established for its key fiber development stages (0–35 DPA). Therefore, it was selected as the core material for spatiotemporal expression analysis of *GbSMXL8* and in vitro ovule culture experiments to ensure the precision and reproducibility of research on fiber development regulatory mechanisms.

Transcriptome analysis revealed that the SMXL gene family exhibited significantly higher expression levels at 0 days post anthesis (DPA) and 35DPA, with *GbSMXL8* (Gbar_A05G03997800), showing marked differential expression across distinct *Gossypium barbadense* cultivars (*p* < 0.01). To further validate the spatiotemporal expression profile of *GbSMXL8*, qRT-PCR was performed to analyze its tissue-specific expression patterns. The results demonstrated that GbSMXL8 was predominantly expressed in roots, 20 DPA fibers, and 35 DPA fibers, with expression levels significantly higher than those in stems and leaves (*p* < 0.05; Figure 4). Notably, the peak expression of *GbSMXL8* coincided with critical stages of fiber development: rapid elongation (20 DPA) and secondary cell wall deposition (35 DPA). These findings suggest that *GbSMXL8* may regulate cotton fiber development through the strigolactone (SL) signaling pathway.

### 2.6. Effect of Auricolactone on Fiber Development

To investigate the role of SLs in cotton fiber development, this study utilized an in vitro ovule culture system of sea island cotton (*Gossypium barbadense*), combined with treatments of the exogenous SL analog GR24 (rac-GR24, 10 μM) and the SL biosynthesis inhibitor Tis108 (5 μM) (Figure 5A) [29,30]. The results showed that, compared to the untreated control group, GR24 significantly promoted fiber elongation (*p* < 0.05), while Tis108 treatment inhibited fiber elongation (*p* < 0.05) (Figure 5B). Further analysis revealed that SL content exhibited a dynamic increase during the rapid fiber elongation stage (20–30 DPA) and was significantly positively correlated with fiber length (*p* < 0.001) (Figure 5C). Notably, GR24 treatment significantly increased ovule weight (*p* < 0.01), whereas Tis108 treatment reduced ovule weight (*p* < 0.05), indicating that SLs promote fiber elongation by modulating material accumulation (Figure 5E,F).

To understand the role of *GbSMXL8* in SL signal transduction, RT-qPCR results showed that the relative expression level of *GbSMXL8* was much higher in GR24 treatment than that in Tis108 treatment at 25 and 30 days of ovule development (Figure 5D). It could be understood that upregulation of *GbSMXL8* gene expression would be triggered when the content of SLs accumulated to a certain extent in vivo. This upregulation could inhibit the tendency of the fibers to elongate. Therefore, we can infer that *GbSMXL8* can respond to the signal changes in SLs to mediate the process of fiber growth and development in cotton.

### 2.7. Impact of Editing GbSMXL8 on Fibers

Based on the known information, we can reasonably speculate that *GbSMXL8*, as a key inhibitor in the SLs signal transduction pathway, could regulate cotton fiber development. In order to further evaluate the specific role of *GbSMXL8* in cotton breeding, especially its potential to improve fiber quality, we planned to edit *GbSMXL8* in *G. hirsutum* varieties with relatively poor fiber quality and analyze its impact on cotton fiber quality. To validate the editing efficiency of CRISPR/Cas9-mediated *GbSMXL8* gene modification and its impact on cotton fiber quality, this study conducted target site mutation analysis on T_0_-generation transgenic plants. Specific primers were designed to amplify the target site and flanking sequences of the *GbSMXL8* gene. PCR products were separated by 1.5% agarose gel electrophoresis and subsequently subjected to sequencing analysis (Figure 5E). Sequencing results revealed that 3 out of 7 positive transformants exhibited target site mutations (editing efficiency: 42.9%*), including single-base insertions (+1 bp) and deletions (−3 bp) (Figure 5B).

This study systematically compared the number and length of stem trichomes between *GbSMXL8* gene-edited transgenic plants and wild-type (WT) plants at the same developmental stage. As shown in Figure 6A, the trichome density of transgenic plants was significantly higher than that of WT (*p* < 0.05). To quantify phenotypic variation, we further measured trichome length for four genotypes (WT, smxl8-1, smxl8-2, and smxl8-3), with three biological replicates per genotype (*n* = 30 trichomes per sample). The results showed:WT: average length of 1.245 ± 0.102 mm (CV = 8.2%), indicating high phenotypic uniformity in WT plants. smxl8-1: average length of 1.744 ± 0.201 mm (CV = 11.5%), representing a 40.1% increase compared to WT (*p* < 0.01), but with a significantly higher coefficient of variation (CV) among samples. smxl8-2: average length of 1.639 ± 0.232 mm (CV = 14.1%), representing a 31.6% increase compared to WT (*p* < 0.05). smxl8-3: average length of 1.516 ± 0.131 mm (CV = 8.6%), representing a 21.8% increase compared to WT (*p* < 0.05) (Figure 7).

These results demonstrate that *GbSMXL8* gene editing not only significantly alters trichome length but also induces phenotypic heterogeneity, which may be related to CRISPR/Cas9 off-target effects or epigenetic regulation.

In addition to changes in length, the stem trichome density of smxl8 transgenic plants increased by 20.8% (smxl8-1), 7.01% (smxl8-2), and 5.26% (smxl8-3) compared to WT. Prediction of cis-acting elements revealed that they were mainly regulated by light. Therefore, photosynthetic performance analysis of *GbSMXL8* transgenic plants showed that chlorophyll content and the Pn and Gs of transgenic plants were significantly increased compared with the WT (*p* < 0.05), indicating that knocking out *GbSMXL8* gene had positive effects on photosynthesis of plants. Meanwhile, the Tr of transgenic plants decreased significantly (*p* < 0.05), further highlighting the important role of *GbSMXL8* in regulating water use and growth of plants (Figure 8).

There were three deletion mutations in the target sequence of *GbSMXL8* mutant cotton. The deletion fragment of target 1 and target 2 was 1–2 bp and 2 bp, respectively. The deletion of the target would lead to frameshift mutations in the amino acid sequence, resulting in phenotypic variants. The t0 plants numbered 1 had a deletion of 1 bp in the target region 1 fragment, a base mutation of 1 bp, and transgenic plants had a longer fiber phenotype (Figure 9A).

To further verify this, we randomly selected T1 cotton fibers with *GbSMXL8* edited, with three samples per group. The average fiber length of the control cotton fiber measured by carding was 23.32 mm, while that of the transgenic cotton increased significantly to 25.97 mm (Figure 9).

## 3. Discussion

*G. hirsutum* and *G. barbadense* are cultivated tetraploid species, domesticated from a common ancestor. Sea island cotton is known as the “king of cotton” with extensive use in high-end textiles because of its long fiber, high strength, and good luster. But the fiber quality of *G. hirsutum* is relatively low, although it has strong adaptability and high yield. Through transgenic technology, *G. hirsutum* can obtain the characteristics of the high-quality fiber of sea island cotton and a highly improved length, strength, and fineness of fiber [2]. There is a close interaction between unicornum lactone (SL) and various plant hormones, including auxins, cytokinins, abscisic acid, jasmonic acid, and GAs. Together with SL, these hormones can control plant growth by forming a complex and integrated regulatory network [7].

In this network, SLs play a key role in effectively increasing the size of the plant meristem and transition zone by inhibiting the transport of auxin so as to accurately regulate and control the elongation process of taproot. At the same time, the lactone of SLs can also inhibit lateral root primordia and its growth, depending on the cytokinin receptor *AHK3* [19]. In addition, the SLs not only has a regulatory effect by itself, but also promotes the synthesis of ethylene, and enhances the transport of auxin and the expression of its receptor *TIR1*, thus further regulating the elongation of root hair [20].

It is noteworthy that elongating cotton fiber is a polarized elongating cell sharing similar regulatory mechanism to root hair development in Arabidopsis [31]. Studies have shown that cotton fibers and stem epidermal hairs have high similarity in molecular regulation mechanism [32]. Several key genes (such as *GhMYB25* and *GhTTG1*) are involved in regulating the development of epidermal hair and cotton fiber at the same time. The functional deletion or abnormal expression of these genes will significantly affect the yield and quality of cotton fiber [33]. For example, deletion of the *GhMYB25* gene not only resulted in a decrease in stem epidermal hair density, but also significantly reduced fiber initiation and ultimately reduced lint percentage [34]. Based on the conservation of this developmental mechanism, the epidermal hair coat of stem and leaf is widely used as a model system for studying the genetic mechanism of cotton fiber development [35]. By comparing the transcriptome data of epidermal hair and fiber development, researchers found that both share core regulatory networks (such as gibberellin and single-leg auractone signaling pathways) in key biological processes such as cell elongation and secondary wall synthesis, which provides important clues for understanding the molecular mechanism of cotton fiber development [36]. In our experiment, the three single-cell models of cotton fiber, root hair, and epidermis exhibited partial regulation conservation in the development process, which could be explained by the regulatory role by similar genes [37]. As an important repressor of SL signaling, *SMXL8* degradation can induce the connection of key transcription factors between SL signaling perception and response. *SMXL8* may also interact with genes in other pathways, such as auxin, Gas, and other hormone signalings to regulate plant growth [36].

Following database-supported identification of 48 SMXL genes, we divided SMXL proteins into four species and into three groups based on similar functions shared by them. Proper functioning of SMXL proteins could be realized according to their specific domains or motifs. Members of the SMXL family have the Clp-N motif, a protein motif that is characteristic of the nucleoside triphosphate hydrolase superfamily and may be responsible for the signaling component of SL sensing. The promoter region of SMXL was analyzed for cis-acting elements. There were widespread cis-elements related to plant growth, especially photo-responsive elementspresent in every SMXL. Hormone-related elements and stress-related cis-elements (e.g., defense and stress-responsive, MYB-related and low-temperature-responsive) were detected in large amounts, providing clues for further exploring SMXL functions [26]. Hence, this experiment used *GbSMXL8* of subgroup III (with the closest evolutionary relationship), which was selected due to its closest evolutionary relationship to other known SMXL genes. The expression of *GbSMXL8* was analyzed in various tissues of *Gossypium barbadense* including fiber, root, stem, and leaf, to uncover its tissue-specific expression patterns. The results revealed that *GbSMXL8* was highly expressed during specific developmental stages of fiber growth, particularly at 20, 25, and 40 days post-anthesis (DPA). This suggests that *GbSMXL8* may play a significant role in fiber development, potentially regulating processes such as cell elongation, secondary cell wall biosynthesis, or other mechanisms critical for fiber maturation.

The fiber length of ovules significantly increased with the rise in strigolactone (SL) content, indicating that SLs play a promoting role in cotton fiber development. To further verify whether *GbSMXL8* is involved in the SL signaling pathway, we examined the expression level of *GbSMXL8* after treatment with the exogenous SL analog GR24. The results showed that the expression of *GbSMXL8* was significantly upregulated following GR24 treatment, confirming our hypothesis that *GbSMXL8* may act as a downstream responsive factor in the SL signaling pathway, participating in the regulation of fiber development. This finding not only supports the important role of SLs in fiber development but also enhances the accuracy of subsequent gene function screening. We constructed the *GbSMXL8* cotton editing vector and found that *GbSMXL8* modulated cotton growth. The phenotype observation of *GbSMXL8* edited plant with an increased number of epidermal hairs indicated that it could interfere with the development process of the cotton epidermal hair. The cis-acting element prediction determined the photosynthetic indexes of the *GbSMXL8* transgenic plants. Transgenic plants had higher chlorophyll content and photosynthetic rate, wherein the Pn represented the accumulation speed of the plants for organic matters.

A higher photosynthetic rate (Pn) indicates that plants can generate more organic matter by utilizing light energy, which promotes favorable growth, development, and stress resistance. Editing the *GbSMXL8* gene has been shown to improve cotton fiber quality. In conclusion, genetic regulation of the *GbSMXL8* gene can enhance cotton fiber quality without adversely affecting normal nutrient metabolism or reproductive growth. This experiment provides theoretical support for subsequent breeding efforts aimed at improving fiber quality in Gossypium hirsutum by utilizing screened superior genes.

The study primarily focused on the functional characterization of *GbSMXL8* in fiber development, but its broader roles in other physiological processes, such as stress responses or nutrient allocation, remain unexplored. Further investigations are needed to determine whether *GbSMXL8* has pleiotropic effects beyond fiber development.

The experiments were conducted under controlled conditions, which may not fully replicate field environments. Field trials are necessary to validate the practical applicability of *GbSMXL8* editing in improving fiber quality under natural growing conditions. The molecular mechanisms by which *GbSMXL8* regulates fiber development, particularly its interactions with other signaling pathways or downstream targets, are not yet fully understood. Additional mechanistic studies, such as transcriptomic or proteomic analyses, would provide deeper insights.

The findings suggest that *GbSMXL8* is a promising target for genetic improvement of cotton fiber quality. However, future studies should explore the potential of combining *GbSMXL8* editing with other fiber-related genes to achieve synergistic effects. Investigating the role of *GbSMXL8* in different cotton species, such as *Gossypium barbadense*, could reveal species-specific regulatory mechanisms and further enhance fiber quality breeding strategies. Long-term studies are needed to assess the stability of fiber quality improvements and any potential unintended effects of *GbSMXL8* editing on cotton agronomic traits. By addressing these limitations and perspectives, future research can build on the current findings to develop more robust and effective strategies for improving cotton fiber quality through genetic regulation.

## 4. Materials and Methods

In this experiment, the materials of the sea island cotton “A130768” of *G. barbadense* were preserved by the Key Laboratory of Agriculture and Biotechnology of Xinjiang Agricultural University, planted in cotton breeding base of Agricultural College of Xinjiang Agricultural University. Materials sampled in this study were petals, stamens, and pistils taken from the day of flowering, as well as young leaves of plants that had grown for about 2 months. All these samples were stored at −70 °C after freezing in liquid nitrogen for later use. *G. hirsutum* “ZM 49” and gene-edited cotton were planted in artificial climate chamber at (26 ± 2) °C with 16 h of illumination.

### 4.1. Structural Analysis of SMXL Gene Family in Cotton

The Cotton FGD database (https://cottonfgd.org/)(accessed on 21 December 2024) was visited to, respectively, download genome-wide translated sequences of *G. arboreum* (A2, CRIv1.0), *G. raymond* (D5, JGI_v2.0), *G. hirsutum* (AD1, ZJUv2.1), and *G. barbadense* (AD2, H7124_ZJUv1.1). Using SMXL protein sequences of Arabidopsis thaliana as a reference, SMXL protein was then searched by local blase tool (E value < 1.0 × 10^−5^). Domain (PF02861) was identified using Hidden Markov Models (HMM) and local HMM searches were performed in the four databases using HMMER 3.0. Cross-screening candidate family genes were screened by these two methods to reduce false positives. Finally, domain analysis on all candidate sequences was completed in InterProScan and SMART after discarding sequences without this domain.

The amino acid residue number, relative molecular weight, hydrophilicity, and theoretical isoelectric point of cotton SMXL protein were analyzed by online ExPASy software (https://web.expasy.org/protparam/) (accessed on 23 December 2024).

### 4.2. Phylogenetic Analysis of Cotton SMXL Family

For this analysis, SMXL protein sequences of six species (Arabidopsis thaliana included) were aligned by ClustalW 2.1 software firstly, followed by the construction of rootless phylogenetic tree by MEGA MEGA 7.0.26. Neighborhood joining method was used to set the number of cycles to 1000.

### 4.3. Gene Structure and Conserved Motif Analysis

SMXL gene structure information was obtained from genome annotation files of *G. hirsutum*, *G. barbadense*, *G. arboreum*, and *G. raymond*, which was then entered into the online Gene Structure Display Server [38] to visually analyze exon/intron structure of SMXL gene. Protein Sequence Retrieval: The protein sequences of SMXL family members from the four cotton species were extracted from the genome annotation files. The protein sequences were submitted to the Pfam database, a widely used resource for protein family analysis. Pfam uses Hidden Markov Models (HMMs) to identify and annotate conserved protein domains. The SMXL protein sequences were analyzed to identify domains such as the Clp-N domain (characteristic of SMXL proteins) and other potential functional motifs. To further visualize the conserved domains, the protein sequences were analyzed using Tbtools (accessed on 28 December 2024) [39], a bioinformatics software that integrates multiple tools for genomic and transcriptomic data analysis. The domain architecture of each SMXL protein was graphically represented, showing the location and arrangement of conserved domains within the protein sequences. This visualization helped clarify the structural and functional conservation of SMXL proteins across the four cotton species.

### 4.4. Cotton SMXL Cis-Acting Element Prediction

The obtained promoter sequences were submitted to the PlantCARE database (http://bioinformations.psb.ugent.be/webtools/plantcare/html/) (accessed on 29 December 2024) for cis-acting element prediction using default parameters (core promoter element threshold Score > 8.0). The predicted results were screened manually to eliminate duplication and low confidence (Score < 5.0) elements, while functional elements related to hormone response, environmental stress, light regulation, and developmental regulation, were retained. The prediction results are visually analyzed by Tbtools software v0.09, and the types, quantities, and distribution positions of each component are counted.

### 4.5. Expression Characterization of the SMXL Family in Cotton

Transcriptome data of fiber development stage of ‘pimas-7″5917’ were retrieved from NCBI website and normalized by Log2 (FPKM + 1). Then, this study constructed the heatmaps using the pheatmap program in R(3.3.0), with the visualization of gene expression patterns simultaneously.

### 4.6. Ovule Culture and Observation

The basal medium was prepared using a specialized cotton ovule in vitro culture medium (PM1671, Coolaber, Beijing, China), consisting of MS basal salts (with vitamin B_5_), 30 g/L glucose, and 0.5 g/L MES buffer. The pH of the medium was adjusted to 5.6 ± 0.1 using 0.1 mol/L NaOH, followed by autoclaving at 121 °C for 20 min (SuoTengYiLiao, STL-FE, Hangzhou, China). After sterilization, the following components were aseptically added under a laminar flow hood according to experimental design: Strigolactone analog: GR24 (Cat. 41012ES03, SESEN, Shanghai, China) at a final concentration of 15 μM. Strigolactone biosynthesis inhibitor: Tis108 (Cat. S25193, Yuanye, Shanghai, China) at a final concentration of 5 Μm [40].

Hormone concentrations were optimized based on pre-experimental trials and referenced to the cotton fiber in vitro culture system established by. The hormone-supplemented medium was aliquoted into sterile glass culture bottles (50 mL per bottle) and stored at 4 °C in the dark until use. Prior to inoculation, the medium was equilibrated to room temperature.

Ovules were collected from sea island cotton (*Gossypium barbadense* cv. ‘A130768’) at 2 days post anthesis (2 DPA). Surface sterilization was performed by immersing ovules in 10% (*w*/*v*) HgCl_2_ solution for 15 min, followed by five rinses with sterile distilled water to remove residual disinfectant. Sterilized ovules were transferred to sterile Petri dishes containing filter paper using pre-chilled sterile forceps.

Sterilized ovules were inoculated into hormone-supplemented medium (20 ovules per bottle) and cultured in a growth chamber under dark conditions at 31 °C and 70% relative humidity for 35 days without agitation to prevent fiber damage. Fiber morphology was monitored at 5, 10, 15, 20, 25, and 30 days of culture using a Leica DM500 optical microscope (10× objective). Fiber length was quantified using ImageJ software (accessed on 2 January 2024) (*n* = 50 fibers per replicate). All experiments included three biological replicates.

### 4.7. Expression Pattern Analysis

A was extracted from cotton tissues (roots, stems, leaves, and fibers at key developmental stages: 5 DPA, 10 DPA, 20 DPA, 25 DPA, and 35 DPA) using the RNA Prep Pure Plant Kit (Cat. No. DP441; Tiangen, Beijing, China). RNA quality was assessed by 1.5% agarose gel electrophoresis, which revealed clear 28S and 18S rRNA bands, indicating high RNA integrity. Subsequently, 1 μg of total RNA was reverse-transcribed into cDNA using the FastKing-RT SuperMix Kit (Cat. No. KR118; Tiangen, Beijing, China) under the following conditions: 42 °C for 15 min and 95 °C for 3 min. The quality of cDNA was verified by PCR amplification of the reference gene *GhUBQ7* (primer sequences: F: 5′-GCTGCTGCTACTGCTACTGC-3′, R: 5′-CAGCTAGCTAGCTAGCTAGC-3′), and the amplification products were confirmed by 1.5% agarose gel electrophoresis.

Specific primers for GbSMXL8 (F: 5′-ATGGCTAGCTACGTAGCTAC-3′, R: 5′-TCAGCTAGCTAGCTAGCTAG-3′) were designed using Primer-BLAST (https://www.ncbi.nlm.nih.gov/tools/primer-blast/) (accessed on 12 December 2024), generating a 150 bp amplicon that spans an intron to avoid genomic DNA contamination. Quantitative real-time PCR (qRT-PCR) was performed in a 20 μL reaction system using SYBR Green I dye, with GhUBQ7 as the internal reference gene. Each experiment included three biological replicates (independent RNA extraction and reverse transcription) and three technical replicates (parallel reactions of the same cDNA template) per sample. The relative expression levels of the target gene were calculated using the ΔCt method, with the formula: ΔCt = Ct(target gene) − Ct(reference gene), and relative expression = 2^(−ΔCt). Data analysis was conducted using one-way analysis of variance (ANOVA) followed by Tukey’s HSD multiple comparison test, with a significance level set at *p* < 0.05.

### 4.8. CRISPR/Cas9 System-Mediated Editing of Cotton

For the *GbSMXL8* gene, according to the sequence of the cotton *GbSMXL8* gene, the online CRISPR2.0 software was used to design editing targets in the first exon region (Figure 1), and the sequences of the two targets were aligned in the cotton genome database to eliminate non-specific target interference.

The plasmid of WMC016 vector was extracted and digested with Bsa I. The sgRNA fragment with target was amplified from the plasmid containing sgRNA vector and ligated with the vector WMC016 after single enzyme digestion. The plasmid was extracted and transformed into Agrobacterium EHA105. To ensure the reliability of transgenic cotton lines, a single bacterial colony harboring the target construct was selected for PCR amplification. Positive clones confirmed by sequencing were subsequently used for Agrobacterium-mediated cotton transformation. The genetic transformation was performed following the protocol established by Ge et al. [41], using the widely cultivated cotton cultivar ‘ZM49’ as the recipient. As a preferred transgenic receptor in China, ‘ZM49’ offers a mature genetic transformation system with an efficiency ≥10% and stable agronomic traits, including consistent fiber quality and yield performance across generations. This high-efficiency system ensures reproducible T_0_ plant generation and minimizes somaclonal variation, making it ideal for functional studies of fiber development-related genes. We used a selectable marker gene (hygromycin resistance) for initial screening. Cotton plants were cultured on medium containing hygromycin (50 mg/L) to select transformants exhibiting antibiotic resistance. The editing efficiency of the target gene (*GbSMXL8*) was confirmed by PCR amplification of the target region, followed by Sanger sequencing. This allowed us to identify specific mutations, including insertions, deletions, or substitutions, in the transgenic lines. Edited lines were compared with non-transgenic controls to assess differences in trichome density, fiber length, and other agronomic traits. After obtaining transgenic plants, seedling DNA was extracted for PCR reaction, with the design of primers according to target sites. Mutations at target sites of positive transgenic plants were detected by sequencing.

### 4.9. Statistics of Fiber and Epidermal Hair Length of Transgenic Cotton

The fiber length of transgenic cotton was measured by combing method. The changes in the number and density of the pelts on the stem and petiole were observed microscopically. The number and length of the pelts on the stem were counted by ImageJ software V 1.54k.

### 4.10. Determination of Physiological and Biochemical Indexes

#### 4.10.1. Measurement of Photosynthetic Rate

To assess the photosynthetic performance of the plants, we employed a portable photosynthesis system (e.g., LI-6800 or similar models), a highly accurate and widely used instrument for measuring photosynthetic parameters in real-time under field or controlled conditions. For each measurement, we selected three leaves of similar growth status to ensure consistency and reliability of the data. These leaves were sampled from closely positioned nodes on the plant to minimize variability due to environmental or developmental differences.

Each photosynthetic indicator was measured three times to ensure data accuracy and reproducibility, and the average value was calculated for further analysis. The key photosynthetic parameters measured included: net photosynthetic rate (Pn), transpiration rate (Tr), stomatal conductance (Gs), and intercellular carbon dioxide concentration (Ci).

#### 4.10.2. Chlorophyll Content Determination

An amount of 0.1 g of the leaf sample was meticulously ground into a fine powder using liquid nitrogen to ensure homogeneous disruption of cellular structures. Subsequently, chlorophyll extraction was performed by adding precisely 1.5 milliliters of 95% ethanol (Sinopharm, 100092683, Beijing, China). The extract was then carefully transferred into a sterile 2-milliliter Eppendorf tube and centrifuged at 12,000 revolutions per minute (rpm) for a duration of 15 min at a controlled temperature of 4 °C to separate the supernatant from cellular debris.

Following centrifugation, the supernatant was meticulously collected, serving as the basis for subsequent chlorophyll content determination. The optical density (OD) values of the supernatant were measured at wavelengths of 665 nanometers (nm) and 649 nm using a spectrophotometer manufactured by UNICO (V1600; Shanghai, China). These specific wavelengths are known to correlate with the absorption peaks of chlorophyll a and b, respectively.

The concentrations of chlorophyll a (Ca) and chlorophyll b (Cb) were then calculated employing the well-established formulas: Ca = 13.95 × OD665 − 6.88 × OD649 and Cb = 24.96 × OD649 − 7.32 × OD665. These calculations were based on the OD values obtained from the spectrophotometric analysis, allowing for the quantitative assessment of chlorophyll content within the leaf samples with a high degree of precision and academic rigor.

### 4.11. Determination of Auricolactone Content

The sample was 0.5 g of the middle part of each leaf’s plant line with uniform growth, and the frozen leaves in liquid nitrogen were tested for auricolactone contentby ELISA [42].

## Figures and Tables

**Figure 1 ijms-26-02293-f001:**
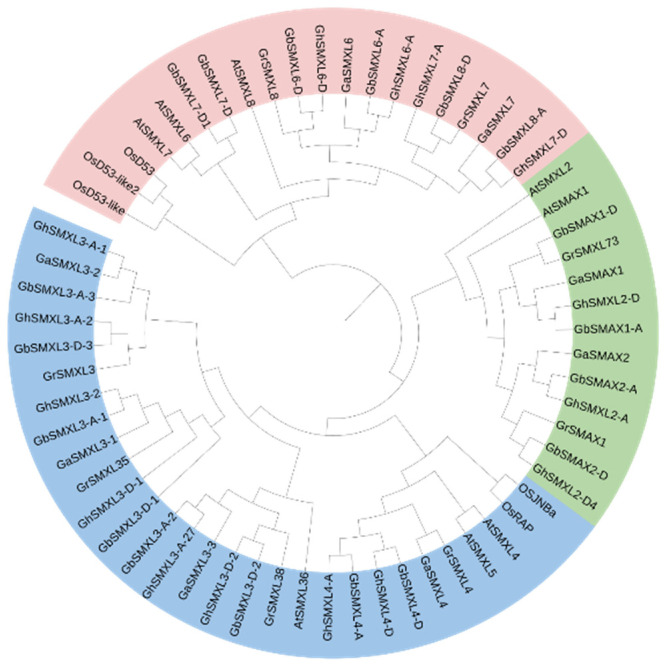
Phylogenetic tree of SMXL protein in *G. arboreum*, *G. barbadense*, *G. raymond*, *G. hirsutum*, *Arabidopsis thaliana*, and *Oryza sativa* L. Alignment of SMXL amino acid sequences was completed using ClustalW, and the phylogenetic tree was constructed by MEGA 7.0 using adjacent linkage method, with 1000 repeats. Gh, Gb, Ga, Gr, Os, and At represent *G. hirsutum*, *G. barbadense*, *G. raymond*, *G. raymond*, *Oryza sativa* L., and *Arabidopsis thaliana*.

**Figure 2 ijms-26-02293-f002:**
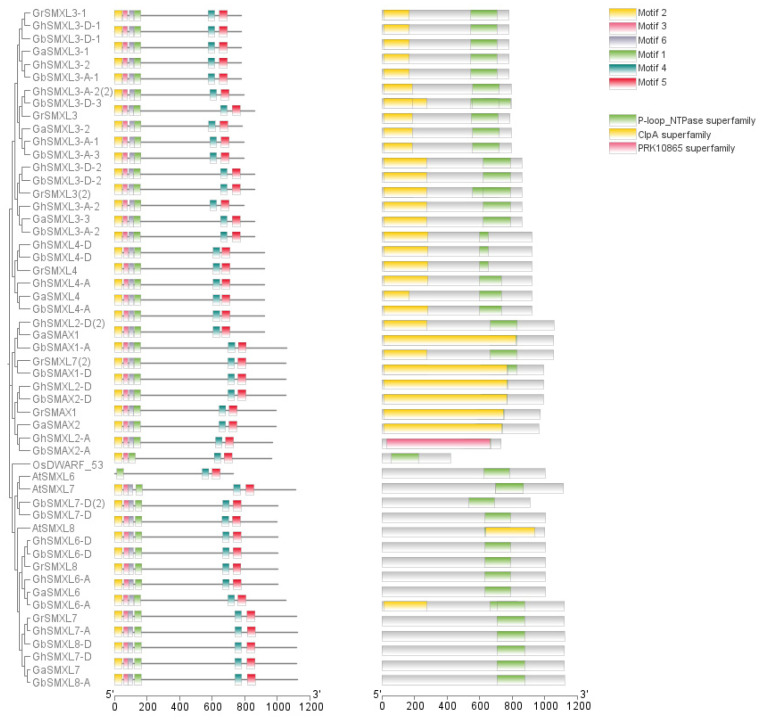
Genetic structure and conserved domains of SMXL protein in four cotton species.

**Figure 3 ijms-26-02293-f003:**
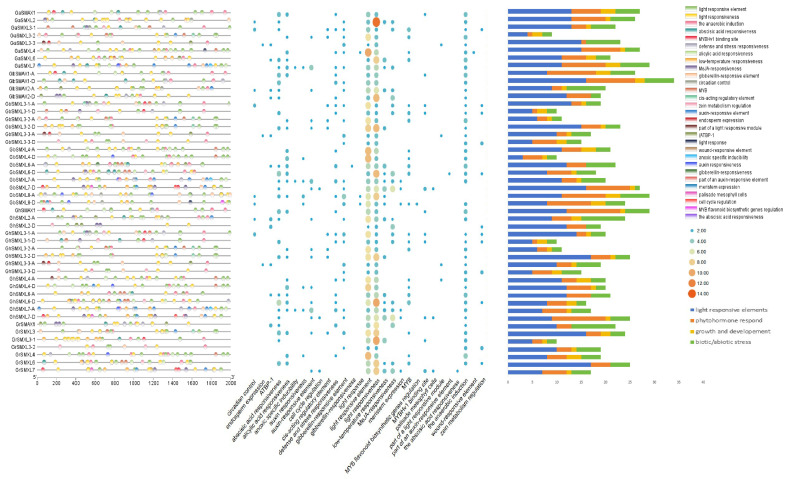
Analysis of cis-acting elements of SMXL in *G. hirsutum*, *G. barbadense*, *G. Raymond*, and *G. arboreum*.

**Figure 4 ijms-26-02293-f004:**
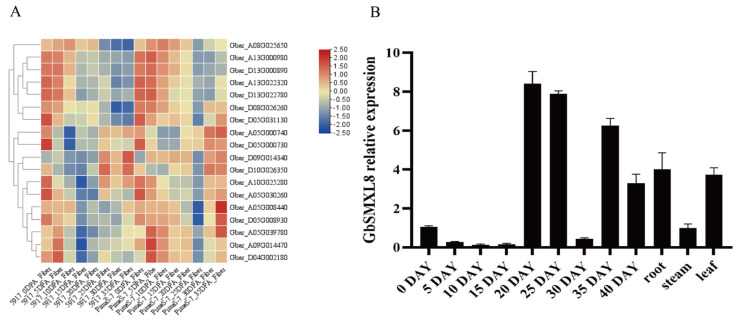
Expression analysis of SMXL during fiber development of sea island cotton. (**A**): Transcriptome expression heat map of different varieties of *G. barbadense*. (**B**): Relative expression level of *GbSMXL8* in different tissues of *G. barbadense*.

**Figure 5 ijms-26-02293-f005:**
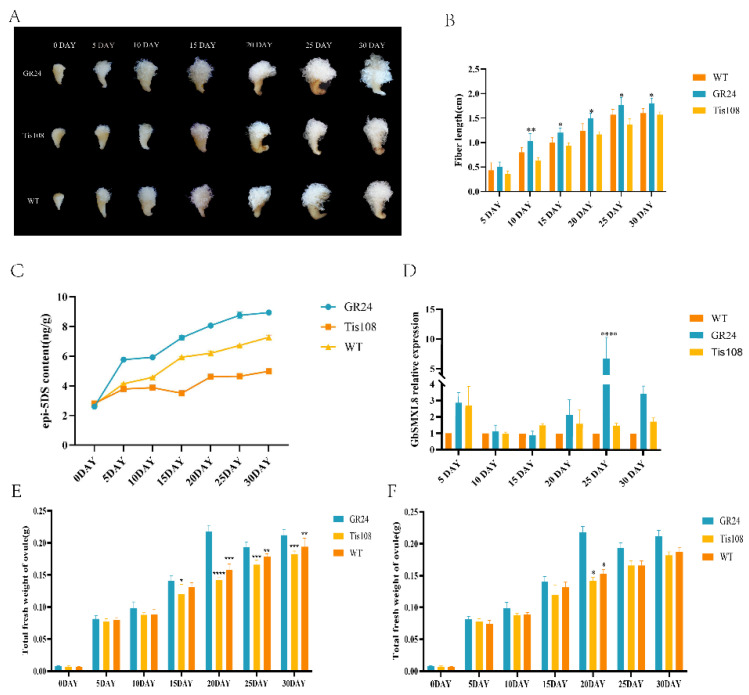
Positive regulation of polygalactone on the fiber development of *G. barbadense*. (**A**): Cotton fiber phenotype (collected at 1 DPA) cultured in vitro for 5, 10, 15, 20, 25, 30 days in a medium containing 15 μM SL synthetic analog GR24, 15 μM SL biosynthesis inhibitor Tis 108, and controls; (**B**): Average fiber length; (**C**): SL epi-5DS content in ovules at varied growth stages; (**D**): relative expression of GbSMXL8 in cotton fibers treated with 15 μM mGR24 or 15 μM Tis 108 for 5, 10, 15, 20, 25, and 30 days; (**E**): total fresh weight of ovule; (**F**): total ovule dry weight. * *p* < 0.05; ** *p* < 0.01; *** *p* < 0.001; **** *p* < 0.0001. Wild type (A130768).

**Figure 6 ijms-26-02293-f006:**
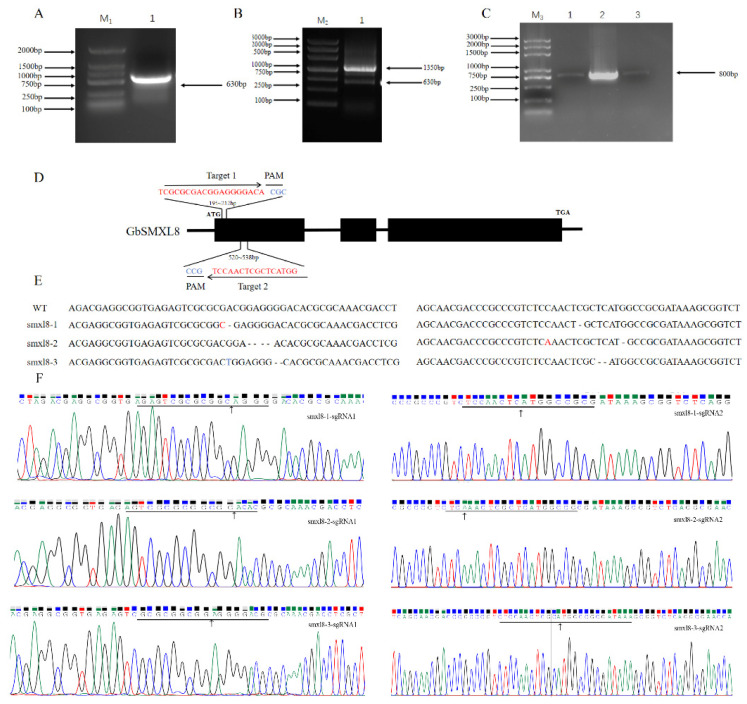
Gene editing SMXL8 target analysis and identification. (**A**): Cloning of double target sgRNA of *GbSMXL8* gene; (**B**): *GbSMXL8* gene; (**C**): PCR product of double target sgRNA Agrobacterium; (**D**): PCR product identification of transgenic plant; (**E**): Two sgRNA positions of *GbSMXL8* (**F**): Sequencing of SmXL8 edited plant.

**Figure 7 ijms-26-02293-f007:**
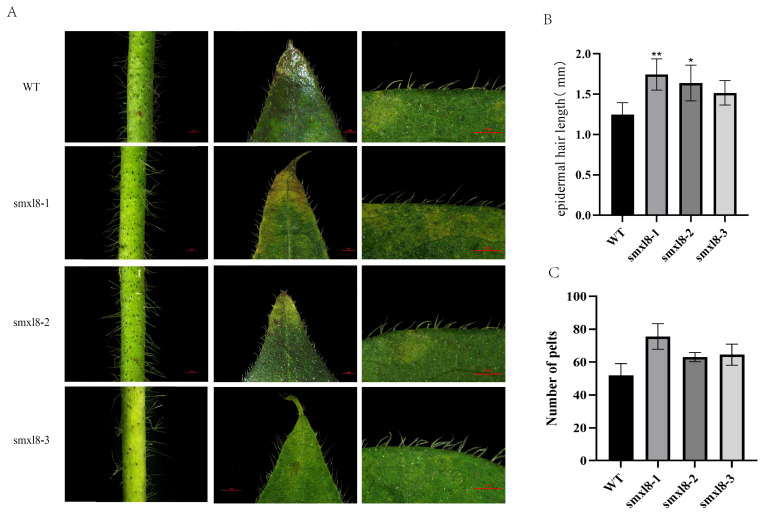
*GbSMXL8* inhibited stem epidermal hair growth (**A**): *GbSMXL8* edited cotton stem and leaf margin epidermal hair; (**B**): *GbSMXL8* edited cotton epidermal hair average length; (**C**): *GbSMXL8* edited cotton epidermal hair number. * *p* < 0.05; ** *p* < 0.01;. WT (ZM 49).

**Figure 8 ijms-26-02293-f008:**
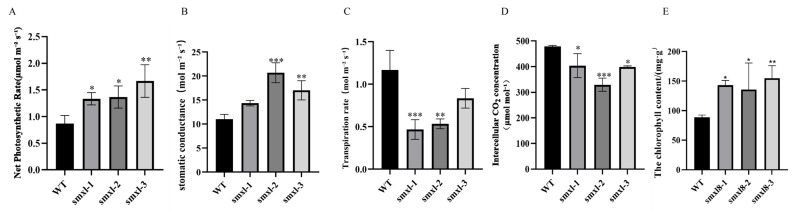
Phenotypic and physiological characterization of transgenic cotton plants. (**A**) Net photosynthetic rate (Pn) in leaves of wild-type (WT) and transgenic cotton lines. (**B**) Stomatal conductance (Gs) in leaves of WT and transgenic lines. (**C**) Transpiration rate (Tr) in leaves of WT and transgenic lines. (**D**) Intercellular CO_2_ concentration (Ci) in leaves of WT and transgenic lines. (**E**) Chlorophyll content in leaves of WT and transgenic lines. Notes:WT: Wild-type control (cultivar ZM 49). Statistical significance: * *p* < 0.05; ** *p* < 0.01; *** *p* < 0.001 (Student’s *t*-test). All data represent mean ± SD (*n* = 3 biological replicates).

**Figure 9 ijms-26-02293-f009:**
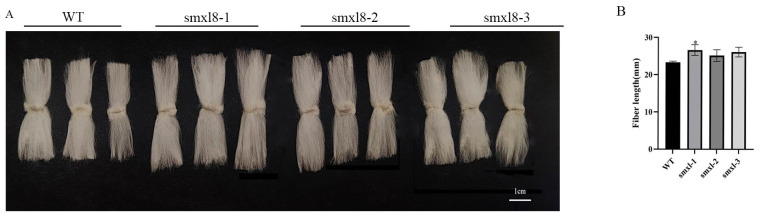
Effect of transgenic plants on fiber development. (**A**): WT, smxl8 transgenic plant fibers; (**B**): smxl8 transgenic cotton leaf mature fiber length. * *p* < 0.05; WT (ZM 49).

**Table 1 ijms-26-02293-t001:** Members of genes from the SMXL family in cotton.

Gene Name	Gene ID	Chromosomal Location	Gene Length (bp)	Number of Amino Acids (aa)	Chromosomal Assignment	Isoelectric Point (IP)	Protein Molecular Weight (kD)
GaSMXL3-1	Ga04G1902	Chr04	2583	860	95242458–95245014	7.692	87.256
GaSMAX1	Ga05G0071	Chr05	3174	1057	748814–752504	8.021	115.686
GaSMXL7	Ga05G3318	Chr05	3294	1097	43831373–43835849	6.859	120.874
GaSMXL2	Ga08G2769	Chr08	3366	1121	127820527–127828153	6.425	123.610
GaSMXL3-2	Ga09G1542	Chr09	2583	860	72503416–72506219	6.833	95.317
GaSMXL6	Ga10G0058	Chr10	3015	1004	617219–621650	8.049	111.242
GaSMXL3-3	Ga13G0098	Chr13	2574	857	977357–979930	6.817	88.262
GaSMXL4	Ga13G2559	Chr10	2769	922	617219–621650	7.633	103.048
GbSMAX1-D	GB_D05G0073	D05	3168	1055	702090–705771	8.024	115.392
GbSMAX2-A	GB_A08G2797	A08	2388	795	119163347–119165732	6.689	89.16
GbSMAX2-D	GB_D08G2782	D08	3168	1055	66175674–66178844	8.024	115.392
GbSMAX1-A	GB_A05G0073	A05	3168	1055	812909–816590	8.024	115.392
GbSMXL3-1-A	GB_A05G4239	A05	2583	860	104306914–104309470	6.833	95.365
GbSMXL3-2-A	GB_A09G1634	A09	2346	781	67281364–67284167	7.939	87.135
GbSMXL3-3-A	GB_A13G0093	A13	2583	795	888559–891131	6.817	88.262
GbSMXL3-1-D	GB_D04G0224	D04	2346	781	2897103–2899659	7.96	87.437
GbSMXL3-2-D	GB_D09G1478	D09	2583	860	42975932–42978737	6.962	95.336
GbSMXL3-3-D	GB_D13G0095	D13	2769	795	744946–747524	6.833	95.365
GbSMXL4-A	GB_A13G2506	A13	2769	922	110467899–110471049	8.124	103.066
GbSMXL4-D	GB_D13G2442	D13	2388	795	58526390–58529536	6.728	89.203
GbSMXL6-A	GB_A10G2794	A10	3018	1005	110924219–110928660	8.286	111.286
GbSMXL6-D	GB_D10G2753	D10	3006	1001	66148734–66153240	8.334	110.726
*GbSMXL7*-A	GB_A05G3145	A05	3351	1116	42056154–42060637	6.784	122.845
*GbSMXL7*-D	GB_D05G3116	D05	3351	1116	33847628–33852136	6.618	122.798
*GbSMXL8*-A	GB_A05G0893	A05	3369	1122	8339616–8343711	6.281	123.759
*GbSMXL8*-D	GB_D05G0872	D05	3369	1122	7368329–7372426	6.395	123.767
GhSMAX1	Gh_A05G005600	A05	3168	1055	874949–879694	8.024	115.441
GhSMXL2-A	Gh_A08G270100	A08	2583	860	123972508–123980097	6.962	95.162
GhSMXL2-D	Gh_D08G260600	D08	2583	860	67120320–67123606	6.962	95.162
GhSMXL3-1-A	Gh_A05G402300	A05	2346	781	106202251–106205180	7.961	87.156
GhSMXL3-2-A	Gh_A09G154300	A09	2583	860	71317475–71320277	6.833	95.33
GhSMXL3-3-A	Gh_A13G010300	A13	2388	795	1013150–1016196	6.751	89.172
GhSMXL3-1-D	Gh_D04G021100	D04	2346	781	2840294–2843087	7.965	87.403
GhSMXL3-2-D	Gh_D09G145500	D09	2583	860	41594390–41597580	6.962	95.162
GhSMXL3-3-D	Gh_D13G010500	D13	2388	795	881251–884228	6.728	89.203
GhSMXL4-A	Gh_A13G235100	A13	2769	922	106078787–106082139	8.124	103.048
GhSMXL6-A	Gh_A10G247300	A10	3006	1001	114136161–114140859	8.334	110.726
GhSMXL6-D	Gh_D10G279400	D10	3018	1005	65267514–65272027	7.945	111.045
GhSMXL7-A	Gh_A05G297200	A05	3315	1116	43110739–43115841	6.784	122.859
GhSMXL7-D	Gh_D05G306200	D05	3351	1116	34015087–34020029	6.618	122.772
GhSMXL4-D	Gh_D13G238200	D13	2769	922	61213860–61217008	7.402	102.95
GrSMXL3-1	Grai_04G022220	Chr04	2583	860	3077802–3080358	6.962	95.293
GrSMXL6	Grai_05G000730	Chr05	3351	1116	113065942–113068498	6.618	122.716
GrSMXL7	Grai_05G033070	Chr05	3366	1121	114074546–114076917	6.507	123.658
GrSMAX1	Grai_08G030620	Chr08	3270	1089	2150749–2153436	7.65	111.174
GrSMXL3	Grai_09G016790	Chr09	2346	781	71185776–71188579	7.574	87.25
GrSMXL6	Grai_10G031250	Chr10	3270	1089	41926634–41929438	7.65	111.174
GrSMXL3-2	Grai_13G001050	Chr13	2388	795	1013713–1016286	6.665	89.293
GrSMXL4	Grai_13G027690	Chr13	2769	922	881673–884251	7.402	103.055

## Data Availability

Data are contained within the article.

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
