# Peer review of "Functional Study of GbSMXL8-Mediated Strigolactone Signaling Pathway in Regulating Cotton Fiber Elongation and Plant Growth"

_ijms, 2025, doi:10.3390/ijms26052293_

Round 1

Reviewer 1 Report

Comments and Suggestions for Authors

Dear Authors,

The aim of the manuscript „Effect of Knock-out of Strigolactones Signal Transducer Gene GbSMXL8 with the CRISPR/Cas9 system on Fiber Development“ was to clarify the role of GbSMXL 8 in cotton fiber development. Also Phylogenetic and cis-element analysis were performed, together with expression analysis. The effect of GbSMXL8-knockout was examined in plant and fiber development, including the response to GR24 in regulating fiber growth. Photosynthetic parameters of the transgenic plants are measured.

Overall, the idea of the manuscript and the experimental setup is good, but the execution and presentation of the results could be significantly improved. In my opinion the title is not appropriate because the effect of the knock-out of strigolactones signal transducer gene is just part of the study. Also, there are repeated words and phrases in the text of the manuscript (line 96, 97, etc); all of the latin names of the plant species and the names of the genes must be in Italic; give the latin names of the plant species, not only the common name – line 95 “rice”, should be “Oryza sativa”, especially because you mention rice for first time in the text.  

The Abstract is very informative and promising comprehensive research. But the rest of the manuscript is very difficult to understand.

The Introduction need to be improved, aims and objectives are missing.

In M&M:

 – You talk about 4 types of cotton used for the analyses - Sea Island Cotton "H7124" and "A130768" of G. barbadense, G. hirsutum and "ZM 49" and gene-edited cotton.

- It is not clear which experiment with what cotton variety was done. You need to specify that.

- The treatment with rac-GR24 and Tis 108 is not described. How you choose the concentrations, what is the brand of the chemicals? Adding detailed information to materials and methods will improve them.

Results:

-        Please provide Figure 3 with better quality.

-        For expression analysis of SMXL during fiber development of Sea Island Cotton, the primer sequences are missing.

-        In 2.6. Effect of auricolactone on fiber development – control is missing (Fig. 5 C, E, F – WT is missing from the graphs).

-        In 2.7. Impact of Editing GbSMXL8 on fibers – avoid discussion in the results part – (line 188-196).

-        It is not clear how you selected the transgenic plants.

-        Fig. 7 – B and C graphs need to be in English; SEM is missing in Fig. 7 C

-        Fig. 8 - it is not clear what is shown, legend is different from what is shown on the figure

The discussion is satisfactory, but is could be expanded and improved.

Reviewer 2 Report

Comments and Suggestions for Authors

The article “Effect of Knock-out of Strigolactones Signal Transducer Gene GbSMXL8 with the CRISPR/Cas9 system on Fiber Development” by Chen et al. addresses investigates the role of the GbSMXL8 gene in cotton fiber development using CRISPR/Cas9 knockout and phylogenetic analysis. The study provides insights into how GbSMXL8 interferes with fiber elongation and plant growth, potentially influencing cotton breeding strategies.

Strigolactones (SLs) play a crucial role in plant development, and understanding their role in fiber growth is important for cotton improvement. The study contributes to this field by exploring the GbSMXL8 gene’s role in SL signaling, but the novelty could be more explicitly articulated. Expanding the discussion on how this research fills a specific gap in SL-mediated fiber growth would strengthen the manuscript.

The study provides valuable data on SMXL gene classification in cotton and its role in fiber development. However, the manuscript should better contextualize its findings within the existing literature, particularly by citing relevant functional studies on SMXL genes in other plant species. A more detailed comparison with previous research would enhance the impact of the study.

The methodology is generally sound but requires some improvements:

- The classification of SMXL genes into groups should be further justified with robust statistical analyses and comparative data from other species;

- The tissue-specific expression analysis should include clearer details on normalization methods, biological replicates, and statistical validation;

- The CRISPR/Cas9 knockout of GbSMXL8 needs further characterization, including sequencing confirmation of mutations, off-target effect analysis, and efficiency assessment.

In addition, the article needs a better structured conclusion that points out the main achievements of the study as well as the possibilities it opens for future research in the area.

The manuscript presents only 4 references from the last five years (2021 onwards), which is incompatible with a topic that has been studied so recently and whose information is renewed at a great speed every day. Also, there are a number of inconsistencies in the references, such as different years in the citation in the text and in the list, and some cited references not included in the list.

This manuscript provides relevant insights into the role of GbSMXL8 in cotton fiber development and SL signaling. However, minor revisions are necessary to improve methodological clarity, strengthen statistical analyses, and enhance literature integration. Addressing these points will significantly enhance the manuscript’s quality and impact for publication in IJMS.

Reviewer 3 Report

Comments and Suggestions for Authors

The authors of the paper "Effect of Knock-out of Strigolactones Signal Transducer Gene GbSMXL8 with the CRISPR/Cas9 system on Fiber Development" describe the effect of the GbSMXL8 gene on cotton fiber development. This research is novel; however, corrections are needed to improve its quality. The introduction has ancient references, so these should be updated, and the citations should be increased. Neither does it present a paragraph detailing the scope of the study, its objective or hypothesis, nor the essential contributions of the research. The results have a weak format; it is necessary to associate them with the research topic and expand the description. The same applies to the discussions, which should be strengthened. On the other hand, it is essential to include a section on conclusions. Specifically, suggestions for corrections are:
References: Cite references in the text following the authors' guide.
Write the scientific names of the plants in italics.
Include a paragraph of studies similar to the one presented (background). 
36-49: The bibliographical references are ancient; they should be updated.
54: The reference is ancient; change it to a recent one.
50-64: Rewrite the paragraph; the ideas are repeated, and it reads redundantly.
67: Include more references; 2012 is not the only one, and it is also 13 years old.
65-72: Include more references for that paragraph.
85-91: Include in that final paragraph the research objective, its importance, scope, and projections.
99: Table 1 is not cited 
Table 1: Do you own the information in the table? If not, include a column with the references.
101-107: Explain why they chose the species mentioned to compare the SMXL relationship. Also, the analysis of the gene structure SMXL is missing; they only describe what is observed in Figure 1, but there is no analysis, and the subheading 2.1 says they do an analysis
Figure 1: Put all the content of the letter in English.
115-122: Rewrite these results. The subheading of this section says that they make an analysis. On the other hand, they mention Figure 3; before that is Figure 2, which is not described.
121,126: Figure 3 has a double description; if both refer to the same thing, unify the information in the same section.
Figure 3: The letters are not legible; they are not readable.
126-139: Include the bibliographic references that support what is described.
143: Write SMXL8 in capital letters.
147-150: The graph in Figure 4 shows a radical decrease in the relative expression of GbSMXL8. You must describe what this effect is attributed to; the text superficially mentions that observation.
158: Make sure that references 14-15 discuss what is mentioned.
160-165: These lines read confusing. Rewrite and improve the technical terms used. Figure 5 contains a lot of information; it is relevant that you provide more depth in describing this result.
Figure 5. Correct the graphs, on the X-axis put Days as the axis title and in the axis dimensioning, put the numbers of those days (i.e., remove the word DAY in each dimensioning). The letters in Figure 5A are not legible. The figure caption is too long; make the information more specific or, if necessary, divide the content into two parts.
169: Conclude the paragraph emphasizing the effect of auricolactone on development.
184-186: Deepen the explanation of the PCR assay and associate it with the cotton fiber quality.
189: Remove the space between the parenthesis of the quotation and the period.
189-191: Mention which genes are being referred to, and add more references to support what has been written.
192-194: Mention which genes are referred to.
200: Remove parentheses
Figure 7: It is not described in the results; what is the purpose of putting this figure? Why is this figure necessary if it is not described?
201-202: If these data are the product of a statistical analysis, incorporate the standard deviations (±).
206, 208: Italicize the P for probability and lowercase it every time you write them.
219: Justify formatting.
Figure 8D: Put CO2 correctly, the 2 should be subscript. 
Figure 8E: Write the Y axis in English.
232, 233: Correctly write p<0.01, p<0.05 and p<0.001
276: What do you mean by IWE?
276-279: Deepen this information and associate the results with what is reported in the literature; it is one of the primary findings in the study.
283: In this line, they indicate that they have a hypothesis regarding SMXL, but it has not been previously described. The introduction must describe this hypothesis, including its scope, the experiments to demonstrate it, and the background that led it to propose it.
293-297: The limitations of the experiments and the perspectives based on the results must be mentioned.
307: To put subscript to the number 2 chemical formula HgCl2.
309: Enter autoclave sterilization conditions, brand of equipment, and manufacturer's data.
309: Insert incubator data, equipment bran,d and manufacturer's data
309: Enter microscope data, make of equipment, and manufacturer's data
317: The specification “e < 1e-10” is not understood.
339-340: Expand the description of this methodology.
350: Put the symbol micro instead of u in microliters.
352: Put the method reference ∆Ct.
368-370: Include the experimental units used, independent repetitions, and statistical program.
379: Put solvent brand and manufacturer's data.
380: What do you mean by the expression "AThe"?
381: Specify the OD symbology.
382: Include the brand and manufacturer's data of the spectrophotometer.
382-384: Expand the description of this methodology; if it is someone's procedure, put that someone's references.
386-388: Detail this methodology, it reads with low technical level.
There are no conclusions, so why do they not present a conclusion paragraph?

Comments on the Quality of English Language

The language needs to be improved, and a thorough technical review is necessary.

Round 2

Reviewer 1 Report

Comments and Suggestions for Authors

Dear Authors,

Thank you very much about your comprehensive answer. But still, there are flaws in the manuscript:

  1. The name of the genes and Latin names of the plant species should be in Italic
  2. Figure 3 is missing
  3. According the plant material, you need to put the information about the selection of the materials in the research in the manuscript. This is the answer of my Comments 1. Also, Suplementary materials and figures are missing
  4. According Comments 7, the steps for transgenic plant screening process are described in detail but only in your answers to the reviewer. There is not such a description in the Materials and Methods section (lines 350-370) as you pointed. Please, add the information in the text of the manuscript.
  5. The new title sounds better and is more appropriate

Reviewer 3 Report

Comments and Suggestions for Authors

Dear authors, your work has improved dramatically; the information is more precise and fluid. Some changes would still enhance the format (see list by line). On the other hand, it is essential that despite describing the conclusions in the discussion section, you incorporate a numeral (it could be 4) to include them there. I mention this recommendation because it is part of the journal's style; however, this suggestion is left to the editor.
60, 62: Put a space between [ ] and :
68: Put a space between teolysis and [16]
73: Put a space between plants and [18]
74: Put a space between perception and [19]
78: Put a space between transduction and [20]
84: Put a space between function and [21,22]
140: Put a space between features: and Clp_A
145: Put a space between processes. and Domain
151: Format as "justified"
173: There is no figure; there is only the figure caption; please add the figure
175: Title 2.5 is incomplete; please restructure it
178: Put the scientific name of the species in italics
183: Figure 3 is cited, but the description refers to Figure 4. Please make sure which figure corresponds to the description.
190, 191, 194, 214, 371: Put the scientific name of the species in italics
218: Remove a; because there are two
240: Put the n in italics
272: Put a space between gene; and B. Put a space between gene; and C
248, 340, 342, 344, 346: Homogenicize the GbSMXL8 throughout the document; this line is not written in italics
340: Remove a period
368: These lines contain the conclusions; although they highlight the relevance of their results, it is necessary to add a numeral called "4. Conclusions"; the articles of this editorial use that style, and it is indicated in the "instructions for authors," please adhere to the guidelines.

Round 3

Reviewer 1 Report

Comments and Suggestions for Authors

Dear Authors,

You have made great efforts and in this way the manuscript is significantly improved

In my opinion, the manuscript is ready for publication now.